# Synthesis of New 1,4-Naphthoquinone Fluorosulfate Derivatives and the Study of Their Biological and Electrochemical Properties

**DOI:** 10.3390/ijms252212245

**Published:** 2024-11-14

**Authors:** Natalia V. Aseeva, Nadezhda V. Danilenko, Evgenii V. Plotnikov, Elena I. Korotkova, Olga I. Lipskikh, Anna N. Solomonenko, Alina V. Erkovich, Daria D. Eskova, Andrei I. Khlebnikov

**Affiliations:** 1Engineering School of Natural Resources, Department of Chemical Engineering, National Research Tomsk Polytechnic University, Lenin Avenue, 30, Tomsk 634034, Russia; eikor@mail.ru (E.I.K.); lipskih-olga@yandex.ru (O.I.L.); ans51@tpu.ru (A.N.S.); avg48@tpu.ru (A.V.E.); 2The School of Advanced Manufacturing Technologies, National Research Tomsk Polytechnic University, Lenin Avenue, 30, Tomsk 634034, Russia; nadezhda.dani@gmail.com; 3Research School of Chemistry & Applied Biomedical Sciences, National Research Tomsk Polytechnic University, Lenin Avenue, 30, Tomsk 634034, Russia; plotnikov.e@mail.ru (E.V.P.); dde5@tpu.ru (D.D.E.)

**Keywords:** 1,4-naphthoquinone fluorosulfate derivatives, SuFEx reaction, anticancer properties, impregnated graphite electrode, voltammetry

## Abstract

This study presents the synthesis of new fluorosulfate derivatives of 1,4-naphthoquinone by the SuFEx reaction. Anticancer properties of obtained compounds were studied on PC-3 (prostate adenocarcinoma), SKOV-3 (ovarian cancer), MCF-7 (breast cancer), and Jurkat cell lines. All the studied compounds showed higher cytotoxic effects than Cisplatin. The DFT method was applied to determine the electronic structure characteristics of 1,4-naphthoquinone derivatives associated with cytotoxicity. A method of determination of 2,3-dichloro-1,4-naphthoquinone (**NQ**), 3-chloro-2-((4-hydroxyphenylamino)-1,4-naphthoquinone (**NQ1**), and 4-((3-chloro-1,4-naphthoquinon-2-yl)amino)phenyl fluorosulfate (**NQS**) in a pharmaceutical substance using an impregnated graphite electrode (IMGE) was developed. The morphology of the IMGE surface was studied using scanning electron microscopy (SEM). The electrochemical behavior of **NQ**, **NQ1**, and **NQS** was studied by cyclic voltammetry (CV) in 0.1 M NaClO_4_ (96% ethanol solution) at pH 4.0 in a potential range from −1 to +1.2 V. Electrochemical redox mechanisms for the investigated compounds were proposed based on the determining main features of the electrochemical processes. Calibration curves were obtained by linear scan voltammetry in the first derivative mode (LSVFD) with the detection limit (LOD) 7.2 × 10^−6^ mol·L^−1^ for **NQ**, 8 × 10^−7^ mol·L^−1^ for **NQ1**, and 8.6 × 10^−8^ mol·L^−1^ for **NQS**, respectively.

## 1. Introduction

A large number of synthetic and natural organic compounds containing a quinone group in their structure can exhibit various biological activities [1]. Many quinones and their derivatives are used as antituberculosis, antimalarial, antimicrobial, antitumor agents, and fungicides [2,3,4,5,6].

The various types of biological activity of 1,4-naphthoquinones, in most cases, are related to the ability of these compounds to take one or two electrons with the formation of the corresponding anion radical (Q^·^^−^) and dianion radical hydroquinone (Q^·^^2−^), as well as to the acid–base properties of these compounds [7,8]. These intermediates interact with essential cellular molecules such as oxygen, DNA, and proteins, altering their biological activity [9,10,11].

The introduction of different functional groups into the structure of 1,4-naphthoquinones affects the biological activity of these compounds. For example, the introduction of nitrogen and/or sulfur atoms into the structure of 1,4-naphthoquinone derivatives has been shown to result in their antifungal, antibacterial, and anticancer activity [12,13,14,15,16]. Since cancer is the second most frequent cause of death in the world, and recent advances in new anticancer agents are very promising, naphthoquinone derivatives are attractive compounds for the development of new anticancer drugs [17,18].

The Sulfur(VI)–Fluoride Exchange (SuFEx) reaction was reported [19] as the next embodiment in the development of Click Chemistry. Nowadays, the SuFEx reaction is successfully used for the synthesis of small molecules, protein labeling, production of polymeric materials, and surface modification [19]. SuFEx click chemistry serves as an effective method for the late-stage functionalization (LSF) of bioactive compounds for rapid diversification of drug-like molecules late in the synthesis process—aiming to enhance both their physical and pharmacokinetic characteristics. Utilizing SO_2_F_2_, Wu’s research team successfully transformed a selection of NIH-approved anticancer drugs into their corresponding fluorosulfate derivatives directly within a 96-well plate [20]. By screening a library of 39 compounds, they identified three aryl fluorosulfates that exhibited improved anticancer activity. Thus, using this reaction, new naphthoquinone derivatives with higher anticancer activity can be obtained.

Therefore, the investigation of biological action mechanisms, as well as the development and synthesis of new compounds with more selective biological activity, requires an understanding of the factors that change the physicochemical properties of quinone systems. The study of the relationship between the redox properties of new compounds and their structure can provide insight for the prediction of 1,4-naphthoquinone (NQ) derivatives’ pharmacological activity. The investigation and description of the redox properties of NQ derivatives is of great importance.

There are a sufficient number of electrochemical techniques for the determination of quinone-containing compounds with cytotoxic properties in the literature (Table 1).

However, since the number of newly synthesized biologically active naphthoquinone derivatives is constantly increasing, new analytical techniques need to be developed.

In the present study, the 1,4-naphtoquinone fluorosulfate derivatives were obtained via the SuFEx reaction, which could be used as a tool for functionalization in order to improve both the physicochemical and pharmacokinetic properties of organic compounds. Anticancer properties of the obtained derivatives were investigated on PC-3 (prostate adenocarcinoma), SKOV-3 (ovarian cancer), MCF-7 (breast cancer), and Jurkat cell lines. An express technique for the quantitative determination of 1,4-naphtoquinone derivatives at the impregnated graphite electrode (IMGE) in a pharmaceutical substance was developed.

Despite the fact that the presented analytical technique is less sensitive compared to the existing methods (Table 1), it can be used for express determination of newly synthesized 1,4-naphtoquinone derivatives. The advantages of using the IMGE in this work are as follows: easy renewal of electrode surface after the adsorption of organic compounds, in contrast to commercial carbon-based electrodes, as well as the absence of expensive and time-consuming modification steps.

## 2. Results and Discussion

### 2.1. The Synthesis of Naphthoquinone Derivatives

One of the investigated compounds, 2,3-dichloro-1,4-naphthoquinone (**NQ**) (with trade names “Dichlone” and “Phygon”), is used as a fungicide for the treatment of fruits, field crops, vegetables, ornamental plants and seeds. Moreover, due to easy accessibility and high stability, **NQ** is known in modern chemistry as a key synthetic precursor for the synthesis of new 1,4-naphthoquinone derivatives with various types of pharmacological activity [31,32].

Naphthoquinone derivatives could be obtained in a number of different ways. Preparation of 1,4-naphthoquinone derivatives from 2,3-dichloro-1,4-naphthoquinone is one of the common methods that can be found in the literature [17,33]. Using this method, we obtained the naphthoquinone derivatives 3-chloro-2-((4-hydroxyphenyl)amino)-1,4-naphthoquinone (**NQ1**) and 3-chloro-2-((3-hydroxyphenyl)amino)-1,4-naphthoquinone (**NQ2**) (Figure 1), which possess antitumor biological activity [34]. The presence of a hydroxyl group makes it possible to further functionalize these compounds through the SuFEx reaction.

Earlier, the use of the SuFEx reaction for the preparation of 1,4-naphthoquinone derivatives led to new potential trypanocidal prototypes against the *Trypanosoma cruzi* (*T. Cruzi*) parasite [35]. In general, SuFEx click reactions provide a useful tool for obtaining bioactive molecules and their late-stage functionalization (LSF) for quick improvements in both physical and pharmacokinetic properties [20].

It is known that aromatic alcohols react selectively with gaseous SO_2_F_2_, leaving aliphatic alcohols, aliphatic and aromatic amines, and carboxylic groups intact [19]. In this view, using the SuFEx reaction between compound **NQ1** and SO_2_F_2_ in the presence of 1,8-diazabicyclo[5.4.0]undec-7-ene (DBU), we obtained 4-((3-chloro-1,4-naphthoquinon-2-yl)amino)phenyl fluorosulfate (**NQS**) (Figure 2). The process was carried out in a two-chamber reactor (Appendix A), where gaseous SO_2_F_2_ was formed in chamber A, while the click reaction proceeded in chamber B.

However, this method gave a very low yield of only 8–9% for compounds **NQS** and **NQS2**. Such yields are not typical for the SuFEx reaction and can be attributed to peculiarities of the naphthoquinone moiety, e.g., to the tendency of naphthoquinones to form stable tautomers [36].

Taking into account the high reactivity of silyl derivatives in SuFEx reactions [19], we have preliminarily synthesized compounds **NQ1a** and **NQ2a** using *tert*-butyldimethylsilyl chloride (TBSCl) according to Figure 3.

We further used the silyl derivatives **NQ1a** and **NQ2a** as substrates for the SuFEx reaction, which was conducted under the same conditions as in Figure 2. The achieved yields of the target products were 95 and 85% for compounds **NQS** and **NQS2**, respectively (Figure 4).

We have checked the possibility of performing a one-pot synthesis of the fluorosulfates **NQS** and **NQS2** without the intermediate isolation of the silylated compounds (Figure 5). Firstly, in chamber B of the reactor, the silylation was carried out, which was controlled by TLC. At the completion of this process, DBU was added to chamber B, while SDI and KF were placed in chamber A of the reactor. The reactor was tightly closed and formic acid was injected through the septum into chamber B to activate the formation of SO_2_F_2_ gas. The one-pot synthesis led to low yields of 32% (**NQS**) and 22% (**NQS2**). Therefore, the isolation of intermediate products **NQ1a** and **NQ2a** is the necessary preparative step of the synthesis.

Thus, we can conclude that the best SuFEx reaction substrates for the synthesis of target naphthoquinone fluorosulfate derivatives are silyl ethers. The reason for this may be the formation of the strong silicon–fluorine bond during the reaction (the Si-F bond dissociation energy is 135 kcal/mol [37]), which easily leads to the formation of the target products.

The spectral and analytical data for the synthesized compounds are given in Materials and Methods. The NMR spectra are shown in Appendix A.

### 2.2. Biological Activity Study

#### 2.2.1. Evaluation of Biological Activity and ADMET Parameters in Silico

The potential biological activity of the synthesized compounds was established using the computer program PASS (Prediction of Activity Spectra for Substances) [38]. The program allows estimating manifestation of different types of biological activity by the compounds with high accuracy and probability. The data are presented in Appendix A, where Pa and Pi values are interpreted as the probabilities of a molecule belonging to the classes of active and inactive compounds, respectively.

According to the PASS data, the investigated naphthoquinones, especially compound **NQ**, may exhibit biological activities against a large number of biotargets. Among them are anticancer effects, which were also discovered previously for different substituted naphthoquinones [39]. Hence, it is reasonable to perform an experimental study of anticancer properties of compounds **NQ**, **NQ1**, **NQ2**, **NQS**, and **NQS2** and evaluate their ADME characteristics.

The ADME properties are necessary in the initial stages of drug development and determine either the bioavailability of a potential drug or its elimination from the study [40]. We assessed the ADME characteristics of compounds **NQ**, **NQ1**, **NQ2**, **NQS**, and **NQS2** using the SwissADME online tool (http://www.swissadme.ch/, accessed on 2 September 2024). According to the results presented in Table 2, the ADME parameters indicate the ability of these five compounds to cross the blood–brain barrier.

We obtained bioavailability radar plots that display drug-likeness scores for the compounds. Six important physicochemical properties were considered, including lipophilicity, molecular size, polarity, solubility, conformational flexibility, and insaturation [41]. The naphthoquinone derivatives studied were found to have generally satisfactory ADME properties, with graphs indicating high bioavailability (Figure 1).

The potential toxicity of the synthesized compounds was established using the Prediction Of Toxicity Of Chemicals (ProTox 3.0) software available online (https://tox.charite.de/protox3/, accessed on 25 August 2024). In silico toxicity assessment showed that compounds **NQ1**, **NQ2**, **NQS**, and **NQS2** belong to toxicity class 4 (Table 2), while compound **NQ** belongs to class 3 (LD_50_ = 160 mg/kg). It should be noted that fluorosulfate derivatives **NQS**, **NQS2** have a higher LD_50_ value; therefore, they are less toxic.

Based on the calculated ADME parameters, LD_50_ values (Table 3), and bioavailability radars (Figure 1), the compounds are expected to have good bioavailability and low toxicity.

#### 2.2.2. The Cytotoxicity Study

It was shown in [42] that sulfur-containing naphthoquinone derivatives showed the greatest activity against the PC-3 prostate cancer cell line. Therefore, the resulting compounds were tested for cytotoxicity (MTT, microscopy) against different cancer cell lines, including PC-3 prostate cancer (Table 3).

A pronounced cytotoxic effect was shown for all the studied compounds (Table 3). Inhibition of tumor cell growth by the investigated naphthoquinones significantly exceeded the effects of the standard cytostatic agent cisplatin. The prostate cancer (PC-3) and lymphoblastic leukemia (Jurkat) cell lines were more susceptible to the effects of the compounds; however, in general, the revealed trends were observed in all studied cell lines. Pairwise comparison of cytotoxicity parameters of compounds **NQ1** and **NQS**, **NQ2** and **NQS2**, respectively, allow us to conclude that compounds **NQ1** and **NQ2** have a slightly higher inhibitory effect on all the studied cell lines in comparison with their fluorosulfate containing analogues. In general, the cytotoxicity parameters of both groups are close. On the other hand, all the modified compounds were found to be significantly superior in cytotoxic effect to the unmodified **NQ** precursor. At the same time, the fluorosulfate group allows modified molecules to be grafted to the surface of materials [43]. This provides a fundamental basis for the development of new hybrid biomaterials.

The breast cancer (MCF-7) cells proved to be relatively more resistant to the effects of all compounds. However, the compounds **NQ2** and **NQS2** showed high cytotoxicity on it; the IC_50_ value was in the range of 4–5 μM, whereas for cisplatin, the IC_50_ was 33.5 μM.

Cell death variants (apoptosis of different stages and necrosis) were assessed on the Jurkat cell line (Figure 2). A dose-dependent increase in the level of cell death was revealed, with apoptosis induction being the main mechanism of this process. The proportion of primary necrotic cells remained low even when exposed to the maximum concentrations of the substances used. The obtained data correlate well with cytotoxicity parameters of studied compounds on other tumor lines.

Massive induction of apoptosis in cell culture is caused by critical unrepairable cell damage after xenobiotic exposure and is usually accompanied by pronounced mitochondrial dysfunction, leading to oxidative stress. This phenomenon was shown by evaluating the level of reactive oxygen species production in cells under the influence of the studied substances (Figure 3).

The level of oxidative stress in cells (the level of ROS production) also correlates with an increase in the concentration of substances (Figure 2). The highest level of induction of oxidative stress in cells was found under the influence of compounds **NQ2** and **NQS2**, which significantly exceeds the standard cytostatic cisplatin in this indicator.

Upon evaluation of the cytotoxic potential of the investigated compounds, **NQ1**, **NQ2**, **NQS**, and **NQS2** demonstrated significant antiproliferative effects across the tested tumor cell lines. The observed cytotoxicity was concomitant with a critical elevation in intracellular oxidative stress parameters and an augmentation of apoptotic markers.

While the cytotoxic response exhibited variability among different tumor cell types, a consistent dose-dependent trend was observed across all experimental cell cultures. This heterogeneity in cell sensitivity may be attributed to the intrinsic differences in cellular redox homeostasis, apoptotic machinery, and drug efflux mechanisms among the diverse neoplastic cell types.

The mechanistic underpinnings of the observed effects suggest that these compounds may act as a redox disrupter, potentially interfering with mitochondrial electron transport and/or inducing the generation of reactive oxygen species. The consequent oxidative stress triggers apoptotic cascades, as evidenced by the increased markers of programmed cell death.

#### 2.2.3. The Electronic Structure of Naphthoquinones in Relation to Cytotoxicity

In this work, we performed quantum chemical calculations using the DFT method for a group of naphthoquinone derivatives to determine the characteristics of their electronic structure related to cytotoxicity. The calculations were carried out using the ORCA 4.2 program. Geometry optimization was performed using the BP86 functional and the triple-zeta basis set def2-TZVPP. Dispersion interactions were accounted for using the D3BJ approximation. This level of theory, according to the literature, provides a good degree of accuracy for the geometry of organic molecules and intermediates [44]. Subsequently, for each compound, calculations were performed without further geometry optimization (single point) using the ωB97X-D3 functional and the 6-311++G(3df,3pd) basis set to obtain more accurate results for orbital energies [45].

For further analysis, we selected the following characteristics of the electronic structure: the energy of the highest occupied molecular orbital [E(HOMO)], the energy of the lowest unoccupied molecular orbital [E(LUMO)], vertical electron affinity (VEA), absolute hardness *η*, absolute electronegativity *χ*, and the reactivity index *ω*. The values of *η*, *χ*, and *ω* were calculated using the equations:*η* = [E(LUMO) − E(HOMO)]/2(1)
*χ* = −[E(LUMO) + E(HOMO)]/2(2)
*ω* = *χ*^2^/(2*η*)(3)

We have found that the energies of frontier molecular orbitals and the related characteristics of the electronic structures of the investigated naphthoquinones (Appendix A) are tightly correlated with IC_50_ values for some of the cancer cell lines (Table 4), with the exception of MCF-7 cells.

Thus, the positive correlations with absolute hardness *η* and electronegativity *χ* show that an increase in these values leads to higher IC_50_, i.e., to lower cytotoxicity with respect to PC-3, SKOV-3, and Jurkat cells. The opposite is true for the E(HOMO) values. According to the correlation data, an increase in the HOMO energy leads to higher cytotoxic activities of the investigated naphthoquinones. The same applies to E(LUMO), although in this case, a significant correlation was obtained only for the PC-3 cells. In general, the DFT results indicate that lower electrophilicity of the compounds would be favorable for the cytotoxic activity.

### 2.3. Electrochemistry

#### 2.3.1. Electrochemical Behavior of NQ, NQ1 and NQS at IMGE

The manifestation of biological activity by naphthoquinones is largely due to their oxidation–reduction properties and processes with electron transfer [46]. In this regard, we carried out a study of the electrochemical behavior of naphthoquinones **NQ**, **NQ1**, and **NQS**. Moreover, using IMGE, it is possible to develop methods for the analysis of biologically active naphthoquinones.

The electrochemical behavior of **NQ**, **NQ1**, and **NQS** was studied by cyclic voltammetry (CV) in 0.1 M NaClO_4_ 96% ethanol solution with the addition of HCl to pH 4.0 in a potential range from −1 to +1.2 V. The analyzed solution contained 0.2 mmol·L^−1^ of the investigated compounds. Figure 4 shows that **NQ**, **NQ1** and **NQS** give a pair of peaks caused by the electrochemical reaction of the quinone group. The introduction of different substituents to the **NQ** structure leads to a shift of the peaks toward more positive values (**NQ1** and **NQS**). Moreover, for **NQ1** and **NQS,** we can note the presence of an additional pair of peaks due to the electrochemical redox reaction of the imino group and one peak (**NQ1**) provided by the electrooxidation of the hydroxyl group.

Table 5 shows the values of the anodic and cathodic peak potentials of the investigated 1,4-naphthoquinone derivatives and their relation with the functional groups.

The cathodic peak corresponding to the quinone group (**NQ**) and the anodic and cathodic peaks corresponding to the imino group (**NQ1** and **NQS**, respectively) were chosen as analytical signals to develop a voltammetric technique for the determination of the compounds.

Two main criteria were applied to confirm the adsorption nature of the electrochemical process. The dependence of the peak current on the potential scan rate should be linear, and the value of the Semerano criterion for the logarithmic dependence of the peak current on the potential scan rate should be greater than 0.5 [47].

As can be seen from Figure 5, for compounds **NQ** and **NQ1,** the dependence of the peak current on the scanning potential rate is linear, whereas for compound **NQS,** this dependence is nonlinear. The values of the Semerano criterion are 0.64, 0.57, and 0.52 for compounds **NQ**, **NQ1**, and **NQS**, respectively. Based on the obtained data, the electrochemical processes for the analyzed compounds were of an adsorption nature.

To study the oxidation–reduction mechanisms of naphthoquinones **NQ**, **NQ1**, and **NQS,** the dependence of the electro reduction current on V^1/2^ was plotted (Figure 6) [48]. This dependence has a nonlinear character for all analyzed compounds. Moreover, the potential difference ΔEp weakly depends on the potential scan rate. All the obtained data point to quasi-reversibility of the electrochemical process for **NQ**, **NQ1** and **NQS** at the IMGE.

It should be noted that we have not obtained any significant correlations between the electrochemical magnitudes and the DFT-derived characteristics (Appendix A). Presumably, the electrode processes are to a great extent determined by adsorption of the compounds on the IMGE surface, which is not directly dependent on the electronic structure of the molecules.

#### 2.3.2. Calculation of the Electron Number

The potential scan rate has great influence on the electrochemical process reversibility. As this rate increases, the process becomes more irreversible. Therefore, at low potential scan rates, it is possible to approximate a quasi-reversible process to a reversible one.

It is necessary to determine the relationship between the potentials of the anodic and cathodic peaks for an electrochemically reversible process to calculate the number of electrons involved in the electrochemical reaction (4) [48]:(4)$$\Delta Ep=2.22\times \frac{RT}{zF}=\frac{0.058}{z}$$

CV with a potential scan rate of 10 mv·s^−1^ was used for the experiment. The number of electrons involved in the electrochemical reaction corresponding to quinone (**NQ**) and imino (**NQ1**) groups was equal to 2, while for **NQS,** just one electron is transferred (imino group).

For all the investigated compounds, the following electrochemical mechanisms were proposed (Figure 7). The mechanism of **NQ** oxidation–reduction is in good agreement with the literature [49].

#### 2.3.3. Voltammetric Determination of NQ, NQ1, and NQS by Linear Scan Voltammetry (LSV)

The experiment revealed the dependence of the **NQ** cathodic signal on pH of the background electrolyte (Figure 8A). HCl or NaOH was added to the background electrolyte to obtain the desired pH value. With the decrease in pH from 6 to 1, the **NQ** cathodic current increased significantly, with its maximum value reached at pH 2.0. With the increase in pH from 6 to 12, the cathodic current values of the **NQ** peak decreased significantly. This fact is explained by the participation of hydrogen ions in the electrochemical reaction, and the facilitation of electroreduction in acidic media [50].

It should be noted that accumulation parameters have a great influence on the analytical signals of the investigated compounds. As the accumulation potential increased, the **NQ** cathodic peak current also increased up to 1 V, followed by a slight decrease (Figure 8B); the accumulation time was 10 s. At the same time, when the accumulation was prolonged from 1 to 100 s at the selected accumulation potential of 1 V, the maximum of **NQ** cathodic peak current was observed at 50 s (Figure 8A).

At the same time, when the pH of the background electrolyte was increased from 2 to 12, the intensity of the **NQ1** anodic peak current and the **NQS** cathodic peak current grew significantly, with the maximum values being reached at pH 10.0 (Figure 9). This fact can be explained by a higher activity of the imino group in an alkaline medium [51,52].

The maximum values of the anodic (**NQ1**) and cathodic (**NQS**) peak currents were obtained at the accumulation potential of −1 V and −1.8 V, and accumulation time of 30 s and 20 s, respectively (Figure 10).

Optimal conditions for the voltammetric determination of **NQ, NQ1,** and **NQS** by linear scan voltammetry in the first derivative mode (LSVFD) at the IMGE are summarized in Table 6.

After selecting the optimal conditions for the LSVFD voltammetric determination of **NQ, NQ1** and **NQS**, the linear calibration plots of the analytical signal on the concentration of the investigated compounds in the model solutions were obtained using LSVFD (Figure 11 A-C).

The pH values are indicated in the figure captions (Figure 11 A–C) and coincide with the data given in Table 6.

Parameters of the LSVFD determination of **NQ, NQ1,** and **NQS** in model solutions are summarized in Table 7.

#### 2.3.4. Interference Study

The developed analytical technique can be used for the determination of newly synthesized 1,4-naphthoquinone derivatives in pharmaceutical preparations, which may contain some excipients. Therefore, it is necessary to determine the selectivity of the developed technique, namely to investigate the interfering effect of such compounds as sodium chloride, lactose, glucose, and sodium acetate.

The presence of interfering components in 10-fold excess reduces current intensity by a maximum of 13% (Figure 12A,B). Consequently, the obtained results show good selectivity in the determination of synthesized 1,4-naphthoquinone derivatives at the IMGE.

#### 2.3.5. Determination of NQ, NQ1, and NQS in Pharmaceutical Substances

The accuracy of the proposed methodology was determined by the “spiked test” method (Table 8).

The results of LSVFD determination of **NQ, NQ1,** and **NQS** in pharmaceutical substances are satisfactory and indicate promising potential for future practical application.

## 3. Materials and Methods

### 3.1. Reagents and Equipment

All the reagents for synthesis were purchased from Sigma-Aldrich (St. Louis, Missouri, USA) or Acros Organics (Pittsburgh, PN, USA) and were used without further purification. LC/MS analysis was performed on an Agilent Infinity chromatograph (Santa Clara, CA, USA) with an Accurate Mass QTOF 6530 mass detector (Santa Clara, CA, USA). Chromatographic conditions: column Zorbax EclipsePlusC18 1.8 μm, 2.1 × 50 mm; eluent H_2_O—acetonitrile (15:85%, *v*/*v*); flow rate 0.2 mL/min. Ionization source: ESI in positive mode. The ^1^H, ^13^C, and ^19^F NMR spectra were recorded on a Bruker AVANCE III HD instrument (operating frequencies: ^1^H—400 MHz; ^13^C—100 MHz; ^19^F—376 MHz). The spectra are presented in the Appendix A. Melting points of the obtained compounds were measured using a Melting Point Apparatus SMP30, heating rate 2.5 °C/min. The reaction mixture was monitored by thin layer chromatography (TLC) on Silufol UV-254 and Merck plates, silica gel 60, F254. Column chromatography was performed using Silica Gel 60 (0.040–0.063 mm).

Sodium hydroxide (NaOH), potassium chloride (KCl) and hydrochloric acid (HCl) for the electrochemical analysis were purchased from Sigma-Aldrich (St. Louis, Missouri, USA) and Sigma Tech (Khimki, Russia). A 0.1 M sodium perchlorate (NaClO_4_) solution was prepared in 96% ethanol and used as the background electrolyte. HCl or NaOH was added to the background electrolyte to obtain the desired pH value. Deionized water was used to prepare all aqueous solutions for the electrochemical experiments. The stock solutions of the analyzed compounds with concentration 0.1 mol·L^−1^ were prepared in dimethylformamide. The solutions were stored for one month in amber bottles at room temperature.

### 3.2. Synthesis of Naphthoquinone Derivatives

3-Chloro-2-((4-hydroxyphenyl)amino)-1,4-naphthoquinone (**NQ1**) and 3-chloro-2-((3-hydroxyphenyl)amino)-1,4-naphthoquinone (**NQ2**) were synthetized as described previously [53,54].

#### 3.2.1. 3-Chloro-2-((4-(tert-butyldimethylsilyloxy)phenyl)amino)-1,4-naphthoquinone (NQ1a)

To a stirred solution of compound **NQ1** (149.9 mg, 0.5 mmol) and imidazole (85.4 mg, 1.25 mmol) in dichloromethane (DCM, 3 mL), a TBSCl solution (113 mg, 0.75 mmol) in DCM (0.5 mL) was added dropwise at 0 °C. The reaction mixture was stirred for 4 h at room temperature, extracted twice with 10 mL DCM, and washed with brine. The organic layer was dried with anhydrous Na_2_SO_4_, filtered, and evaporated in vacuo. The residue was crystallized from ethanol. Yield 99%. M.p. 191–192 °C. ^1^H NMR (CDCl_3_), δ, ppm: 8.18 (1H, d, *J* = 8 Hz, H-4), 8.10 (1H, d, *J* = 8 Hz, H-1), 7.76 (1H, td, *J* = 8 Hz, 1.3 Hz, H-3), 7.67 (1H, td, *J* = 8 Hz, 1.3 Hz, H-2), 7.64 (1H, s, NH), 6.99 (2H, d, *J* = 8 Hz, H-6, H-7), 6.81 (2H, d, *J* = 8 Hz, H-5, H-8), 0.99 (9H, s, *t*-Bu), 0.21 (6H, s, SCH_3_). The atom numbering used for the ^1^H NMR signal assignments are shown in Figure 13. ^13^C NMR (CDCl_3_), δ, ppm: 180.6, 177.5, 154.0, 141.9, 135.0, 132.8, 132.7, 130.9, 129.8, 127.1, 126.9, 126.3, 119.9, 113.5, 25.7, 18.3, −4.4. Found, %: C, 63.55; H, 5.72; N, 3.51; C_22_H_24_ClNO_3_Si. Calculated, %: C, 63.83; H, 5.84; N, 3.38.

#### 3.2.2. 3-Chloro-2-((3-(tert-butyldimethylsilyloxy)phenyl)amino)-1,4-naphthoquinone (NQ2a)

The synthesis of **NQ2a** was carried out as described above for obtaining compound **NQ1a**, using **NQ2** as a starting material. Yield 96%. M.p. 188–189 °C. ^1^H NMR (CDCl_3_), δ, ppm: 8.19 (1H, d, *J* = 8 Hz, H-4), 8.11 (1H, d, *J* = 8 Hz, H-1), 7.77 (1H, t, *J* = 8 Hz, H-3), 7.69 (1H, t, *J* = 8 Hz, H-2), 7.63 (1H, s, NH), 7.18 (1H, t, *J* = 8 Hz, H-7), 6.70 (2H, d, *J* = 8 Hz, H-6, H-8), 6.57 (1H, s, H-5), 0.98 (9H, s, *t*-Bu), 0.21 (6H, s, SCH_3_). ^13^C NMR (DMSO-d_6_), δ, ppm: 180.5, 177.2, 155.3, 143.7, 140.5, 135.2, 133.7, 132.4, 130.8, 129.1, 127.0, 126.6, 117.8, 116.5, 116.0, 115.1, 26.0, 18.4, −4.0. Found, %: C, 63.76; H, 5.68; N, 3.49; C_22_H_24_ClNO_3_Si. Calculated, %: C, 63.83; H, 5.84; N, 3.38.

#### 3.2.3. 4-((3-Chloro-1,4-naphthoquinon-2-yl)amino)phenyl fluorosulfate (NQS)

Chamber A of a two-chamber reactor (Appendix A) was filled with 1,1′-sulfonyldiimidazole (SDI, 495 mg, 2.5 mmol) and potassium fluoride (378 mg, 6.5 mmol). Next, chamber B was charged with compound **NQ1a** (207 mg, 0.5 mmol), DBU (75 µL, 0.5 mmol), and DCM (3 mL). Finally, 1.6 mL formic acid was added by injection through the septum in the cap of chamber A, and instant gas formation was observed. After 24 h stirring at room temperature, one of the caps was carefully opened to release the residual pressure. The reaction mixture was stirred for another 15 min to ensure that all sulfuryl fluoride was removed. Next, the content of chamber B was transferred to a 100 mL round-bottomed flask. Chamber B was rinsed two times with 2 mL of DCM, and these fractions were added to the same flask. Then, the solvent was removed under reduced pressure. The crude product was purified by column chromatography on silica gel (eluent chloroform). Compound **NQS** was obtained as orange crystals. Yield 95%. M.p. 213–214 °C. ^1^H NMR (DMSO-d_6_), δ, ppm: 9.46 (1H, s, NH), 8.04 (2H, d, *J* = 8 Hz, H-1, H-4), 7.87 (1H, td, *J* = 7, Hz 1.2 Hz, H-3(2)), 7.82 (1H, td, *J* = 7 Hz, 1.2 Hz, H-2(3)), 7.53 (2H, d, *J* = 9 Hz, H-6, H-7), 7.28 (2H, d, *J* = 9 Hz, H-5, H-8). ^13^C NMR (DMSO-d_6_), δ, ppm: 180.4, 177.4, 145.9, 143.5, 140.3, 135.2, 133.9, 132.3, 130.9, 127.1, 126.6, 125.3, 121.2, 117.0. ^19^F NMR (DMSO-d_6_), δ, ppm: 38.34. LC/MS (ESI+); m/z: 381.9945 [M + H]^+^ experimental ([C_16_H_9_ClFNO_5_S + H]^+^ = 381.9947 theor.).

The yield of **NQS** was 8% when using compound **NQ1** as a starting material.

#### 3.2.4. 3-((3-Chloro-1,4-naphthoquinon-2-yl)amino)phenyl fluorosulfate (NQS2)

The synthesis of **NQS2** was carried out as described above for obtaining compound **NQS**, using compound **NQ2a** as a starting material. Yield 85%. M.p. 198–200 °C. ^1^H NMR (DMSO-d_6_), δ, ppm: 9.52 (1H, s, NH), 8.05 (2H, d, *J* = 8 Hz, H-1, H-4), 7.88 (1H, td, *J* = 7 Hz, 1.2 Hz, H-3(2)) 7.83 (1H, td, *J* = 7 Hz, 1.2 Hz, H-2(3)), 7.50 (1H, t, *J* = 8 Hz, H-7), 7.33 (1H, s, H-5), 7.25–7.30 (2H, m, H-6, H-8),. ^13^C NMR (DMSO-d_6_), δ, ppm: 180.4, 177.5, 149.6, 143.3, 141.9, 135.2, 133.9, 132.2, 130.9, 130.4, 127.0, 126.6, 123.7, 118.0, 116.2, 115.8. ^19^F NMR (DMSO-d_6_), δ, ppm: 38.84. LC/MS (ESI+); m/z: 381.9947 [M + H]^+^ experimental ([C_16_H_9_ClFNO_5_S + H]^+^ = 381.9947 theor.).

The yield of **NQS2** was 9% when using compound **NQ2** as a starting material.

#### 3.2.5. Method of One-Pot Synthesis 4-((3-Chloro-1,4-naphthoquinon-2-yl)amino)phenyl fluorosulfate (NQS) and 3-((3-Chloro-1,4-naphthoquinon-2-yl)amino)phenyl fluorosulfate (NQS2)

Chamber A of a two-chamber reactor (Appendix A) was filled with 1,1′-sulfonyldiimidazole (SDI, 495 mg, 2.5 mmol) and potassium fluoride (378 mg, 6.5 mmol). Next, chamber B was charged with compound **NQ1** (149.9 mg, 0.5 mmol), imidazole (85.4 mg, 1.25 mmol) in dichloromethane (DCM, 3 mL), and a TBSCl solution (113 mg, 0.75 mmol) in DCM (0.5 mL) was added dropwise at 0 °C. The reaction mixture in chamber B was stirred for 4 h at room temperature (the presence of **NQ1** was controlled by TLC in CHCl_3_). After finishing reaction, a DBU (75 µL, 0.5 mmol) was added in chamber B, and both reactor caps were tightly closed. Finally, 1.6 mL formic acid was added by injection through the septum in the cap of chamber A, and instant gas formation was observed. After 24 h stirring at room temperature, one of the caps was carefully opened to release the residual pressure. The reaction mixture was stirred for another 15 min to ensure that all sulfuryl fluoride was removed. Next, the content of chamber B was transferred to a 100 mL round-bottom flask. Chamber B was rinsed two times with 2 mL of DCM, and these fractions were added to the same flask. Then, the solvent was removed under reduced pressure. The crude product **NQS** was purified by column chromatography on silica gel (eluent chloroform). Yield 32%. Spectra as described above.

The synthesis of **NQS2** was carried out the same way using compound **NQ2** as a starting material. Yield 22%. Spectra as described above.

### 3.3. Biological Experiments

*Cell line preparation.* PC-3 (prostatic adenocarcinoma), SKOV-3 (ovarian cancer), MCF-7 (breast cancer) cell lines (PrimeBioMed LLC, Moscow, Russia) were grown in Dulbecco’s modified eagle medium (DMEM, Gibco, Billings, MT, USA). Cultivation was carried out under standard conditions in a CO_2_-incubator CB-170 (Binder, Germany) with 5% CO_2_ at 37 °C and 100% humidity. The cells were thus brought into the stable growth phase and were used for the experiments.

*Cell viability tests.* To evaluate the cytotoxic effects, 5000 cells (counted by automated cell counter Countess 2FL (Thermo Fisher Scientific Inc., Waltham, MA, USA) per well were seeded into wells of a 96-well sterile plate (SPL Life Sciences Co., Naechon-myeon, Republic of Korea) at 24 h before the start of the tests. The cells adhered and adapted to the substrate during this period. Following this, the medium was replaced with fresh medium containing serial dilutions of the test compounds to evaluate their effects. Media contained 50 μM, 25 μM, 12.5 μM, 6.25 μM, 3.13 μM, 1.56 μM, 0.78 μM, 0.39 μM concentration for each tested compound. All plates were placed in an incubator and cultured under standard conditions at 37 °C and 5% CO_2_.

To assess cell viability, MTT assays were employed after a 24-h incubation period. Specifically, the DMEM medium in the wells was replaced with a 0.45 mg/mL solution of 3-(4,5-dimethylthiazol-2-yl)-2,5-diphenyl-2*H*-tetrazolium bromide (MTT, PanEco Ltd., Moscow, Russia). The plate was then incubated in a CO_2_-incubator at 37 °C for 4 h. Following incubation, the MTT solution was aspirated, and 100 µL of dimethyl sulfoxide (DMSO, PanEco Ltd., Moscow, Russia) was added to solubilize the formazan crystals. The optical density of the resulting solution was measured at 570 nm using a Multiskan FC microplate photometer (Thermo Fisher Scientific Inc., Waltham, MA, USA). Data were analyzed, and cell viability was expressed as a percentage relative to the control group. An extended viability assessment was conducted using the Jurkat cell line. For this, 40,000 cells per well, counted via the automated cell counter Countess 2FL (Thermo Fisher Scientific Inc., Waltham, MA, USA), were seeded into a sterile 96-well plate (SPL Life Sciences Co., Naechon-myeon, Republic of Korea). All compounds were added in serial dilutions at identical concentrations. After 24 h of incubation, cytofluorimetric measurement (CytoFlex, Beckman Coulter, Brea, CA, USA) of cell death variants, cell viability and the level of reactive oxygen species (ROS) production was performed with Annexin V-FITC Apoptosis Detection Kit (Abcam, Cambridge, UK and Cellular ROS Assay Kit (Abcam, UK) according to manufacturer’s protocol. All the tests were performed in 6 replications.

### 3.4. DFT Calculations

The structures of the studied naphthoquinones were constructed and preliminarily optimized using the semi-empirical PM3 method with the HyperChem 7 software (Hypercube, Inc., Gainesville, FL, USA). DFT calculations were performed using the ORCA 4.2 program (Max Planck Institute for Coal Research, Mülheim/Ruhr, Germany, December 2018) on a computer running Windows Server 2016 (CPU 16 × 2.4 GHz, 16 GB RAM). The BP86 functional [55], the def2-TZVPP basis set [56], the RI approximation with the auxiliary basis set def2/J [57], and the D3BJ dispersion correction [58] were used. The attainment of real energy minima during geometry optimization in the gas phase was confirmed by calculating the frequencies of normal vibrations. Subsequently, for the energy minimum found for each compound, a single point energy calculation was performed without geometry optimization using the ωB97X-D3 functional [45] and the 6_−_311++G(3df,3pd) basis set with the RIJCOSX approximation [59,60], along with auxiliary basis sets generated by the AutoAux procedure [57]. To evaluate the vertical electron affinity (VEA), “single point” energies of anion radicals were calculated, whose geometries corresponded to the optimized structures of the respective neutral naphthoquinones. The ωB97X-D3 functional were used for these calculations, as described above. The obtained results were visualized and analyzed using the Chemcraft program (https://www.chemcraftprog.com, accessed on 20 June 2024).

### 3.5. Electrochemical Measurements

#### 3.5.1. Equipment

Cyclic voltammetry (CV) and linear scan voltammetry in the first derivative mode (LSVFD) were performed with TA-Lab voltammetric analyzer in a conventional three-electrode cell (Tomanalyt, Tomsk, Russia). IMGE was used as a working electrode, silver/silver chloride electrodes (1 mol·L^−1^ KCl) were used as counter and reference electrodes. The analyzer Itan (pH-meter/ionomer) was used for pH measurement. All the experiments were carried out at room temperature. The surface of the IMGE was examined using a JEOL JSM-7500FA scanning electron microscope (SEM) operated at an accelerating voltage of 10 kV and equipped with a field emission cathode and a secondary electron (SE) detector.

#### 3.5.2. Characterization of the IMGE

IMGE represents a disk graphite electrode impregnated with polyethylene and paraffin under vacuum [61]. The general SEM image obtained for the electrode surface is shown in Figure 14A.

The electrode material is composite. This means that it retains micro porosity due to the nonregular distribution of polyethylene and paraffin (phase) during the impregnation procedure. Figure 14B shows traces of impregnate (a mixture of polyethylene and paraffin), as well as scratches arising from mechanical cleaning of the IMGE surface on the ashless filter. Moreover, residual pores from the primary graphite blank also appear on the surface of the IMGE.

The presence of micropores with a diameter of about 2 μm surrounded by individual pores with a smaller diameter of about 200 nm is observed under sufficient magnification (Figure 14B). Consequently, it is possible to assume that an adsorption process on the electrode surface plays a sufficient role during electrochemical measurements.

The redox pair [Fe(CN)_6_]^3−^/ [Fe(CN)_6_]^4−^ is widely used for evaluation of the IMGE efficiency as a working electrode. Figure 15 shows cyclic voltammograms of [Fe(CN)_6_]^3−^/ [Fe(CN)_6_]^4−^ at the IMGE. CV was used to calculate the value of the IMGE electroactive surface area, namely, the redox current of [Fe(CN)_6_]^3−^/ [Fe(CN)_6_]^4−^ was recorded with the potential scan rate from 20 to 200 mV·s^−1^.

The value of electroactive surface area of IMGE was estimated as 0.059 cm^2^ using the Randles–Ševčík equation [62].

Electrochemical properties of the electrode were also evaluated through impedance spectroscopy (EIS). Figure 16A shows the EIS spectra of the IMGE in the capacity coordinates in the phosphate buffer solution (PBS), pH 6.86. The resulting spectrum corresponds to an equivalent electrical circuit (Figure 16A inset). The accuracy of the selected scheme is confirmed by the small error values of the simulated circuit elements, less than 5% and by the value of the criterion χ^2^ = 5.7 × 10^−3^. Double layer capacitance (C_dl_) value was calculated as 2.148 µF [63].

Only solvated ions, which are responsible for the presence of the double layer capacitance, are present in PBS. When the [Fe(CN)_6_]^3−^/[Fe(CN)_6_]^4−^ probe is applied, a component appears that passes through the double layer and can diffuse near the electrode surface. The impedance spectrum of the IMGE in a solution of 5·10^−3^ mol·L^−1^ [Fe(CN)_6_]^3−^/[Fe(CN)_6_]^4−^ in 0.1 mol·L^−1^ KCl in Nyquist coordinates is shown in Figure 16B. The resulting spectrum corresponds to an equivalent electrical circuit (Figure 16B inset). The value of the criterion χ^2^ = 4 × 10^−4^; the R_ct_ value is 6.94 kΩ and is acceptable for measurements at IMGE.

The obtained values of the double layer capacitance and charge transfer resistance indicate that the oxidation and reduction processes at the IMGE occur without any difficulties. Therefore, it can be anticipated that other compounds will be easily oxidized and reduced on this surface.

### 3.6. ADME Predictions

The ADME and physicochemical properties of selected compounds were computed using SwissADME (http://www.swissadme.ch, accessed on 2 September 2024).

## 4. Conclusions

This work reported for the first time the preparation of fluorosulfate derivatives of naphthoquinone. It was shown that these compounds had cytotoxic activities against PC-3 (prostatic adenocarcinoma), SKOV-3 (ovarian cancer), MCF-7 (breast cancer) cell lines and possess satisfactory bioavailability and toxicity parameters. The parameters of apoptosis, cell viability and the level of production of reactive oxygen species (ROS) were studied on the Jurkat cell line. A voltammetric technique for the determination of 1,4-naphthoquinone derivatives at the IMGE was developed. As a result of the studies conducted with the use of IMGE, we established the nature of the oxidation–reduction processes of the analyzed compounds **NQ**, **NQ1**, and **NQS**. Namely, for all the analytes, the process was characterized as quasi-reversible with the participation of two electrons for **NQ** and **NQ1** and one electron for **NQS**.

The LSVFD was used to obtain the calibration curves of the analytical signal vs. concentration of the investigated compounds under the optimal experimental conditions with LOD 7.2 × 10^−6^ mol·L^−1^ for **NQ**, 8 × 10^−7^ mol·L^−1^ for **NQ1**, and 8.6 × 10^−8^ mol·L^−1^ for **NQS**.

The proposed technique was successfully applied for the determination of compounds in the substance by the spiked-test method. The use of IMGE provides a promising opportunity not only to investigate the nature of electrochemical reactions of 1,4-naphthoquinones, but also to use a simple and rapid technique for the determination of the naphthoquinone derivatives both in pharmaceutical substances and in real objects.

## Data Availability

The data presented in this study are available in this article.

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
