# Peer review of "Synthesis of New 1,4-Naphthoquinone Fluorosulfate Derivatives and the Study of Their Biological and Electrochemical Properties"

_ijms, 2024, doi:10.3390/ijms252212245_

Round 1
Reviewer 1 Report
Comments and Suggestions for Authors
The study presents a comprehensive exploration of new fluorosulfate derivatives of 1,4-naphthoquinone synthesized through the SuFEx reaction, and their promising anticancer properties, which exceed those of Cisplatin, across various cancer cell lines. The proposed redox mechanisms and the calibration data further enhance the paper's relevance to both pharmaceutical and electrochemical applications. However, some minor revisions are necessary to ensure the paper's clarity and completeness.
1- Figure 1, which appears to focus on radar bioavailability, is not explained in the materials and methods section.
2-It is important to include the number of replications for the experiments reported in Table 3, Table 4, and Figures 2 and 3, as this information is crucial for assessing the statistical validity and reproducibility of the results.
3. At table 3 the unit of concentration was missing
4. Cell source needs to be added to the material and method section
5- Details regarding the acquisition of SEM images, which is currently missing from the methods section.
Author Response
Comment 1: Figure 1, which appears to focus on radar bioavailability, is not explained in the materials and methods section.
Response 1: We are thankful for the important remark provided by the reviewer. According to this remark, we have made the necessary changes and added the “3.6. ADME Predictions” section in Materials and Methods.
Comment 2: It is important to include the number of replications for the experiments reported in Table 3, Table 4, and Figures 2 and 3, as this information is crucial for assessing the statistical validity and reproducibility of the results.
Response 2: According to the reviewer’s recommendation we have changed Table 3 and Figures 2 and 3. Table 4 should not contain the number of repetitions because these are correlation coefficients.
Comment 3: At table 3 the unit of concentration was missing.
Response 3: Following the recommendation of the reviewer, we added the concentration unit (μM) to the title of Table 3.
Comment 4: Cell source needs to be added to the material and method section.
Response 4: According to the reviewer’s recommendation, we have added this information in revised version of the manuscript.
Comment 5: Details regarding the acquisition of SEM images, which is currently missing from the methods section.
Response 5: According to the reviewer’s recommendation, we have added this information in the Materials and Methods section
Reviewer 2 Report
Comments and Suggestions for Authors
The manuscript is a significant contribution to the field of fluorosulfate derivatives of naphthoquinones. The authors discussed the synthesis and biological activity of the 1,4-naphthoquinone derivatives. The manuscript is well-written and summarized. This reviewer has some suggestions.
1. The authors showed the synthesis of fluorosulfate derivatives of 1,4-naphthoquinones using the SuFEx reaction. Although the reaction is reported in the literature, the authors showed that silicon ether compounds are the best substrate for the SuFEx reaction. A similar reaction is reported in the literature with phenol substrate with excellent yields. What is the reason for the low yields for the SuFEX reaction of 1,4-naphthoquinones?
2. The authors showed that silicon ether substrate gave up to 95% yields in the SuFEx reaction. The authors are asked to provide a plausible reaction mechanism and some mechanistic study of why the phenol substrate provided lower yields.
3. Authors discussed the predicted pharmacokinetics and other ADMET properties of 1,4-naphthoquinones. Can some real ADMET properties and pharmacokinetics of a few potent compounds be provided to show the drug-likeness of the compounds?
4. In the experimental section, there is no mention of MHz in 1H NMR. Please provide. In the 13NMR, it is mentioned that 400 MHz. Is it performed on a 1600 MHz machine? Please correct it. NMRs are not written properly. The format is not correct. Please check it.
Author Response
Comment 1: The authors showed the synthesis of fluorosulfate derivatives of 1,4-naphthoquinones using the SuFEx reaction. Although the reaction is reported in the literature, the authors showed that silicon ether compounds are the best substrate for the SuFEx reaction. A similar reaction is reported in the literature with phenol substrate with excellent yields. What is the reason for the low yields for the SuFEX reaction of 1,4-naphthoquinones?
Response 1: The authors thank the reviewer for this important question. We addressed it in the manuscript: “Such yields are not typical for SuFEx reaction and can be attributed to peculiarities of the naphthoquinone moiety, e.g. to the tendency of naphthoquinones to form stable tautomers [36].” Also, regarding the higher yields obtained with the silyl derivatives we wrote: “The reason for this may be the formation of the strong silicon-fluorine bond during the reaction (the Si-F bond dissociation energy is 135 kcal/mol [37]), which easily leads to the formation of the target products.” But, of course, these hypotheses require a more in-depth study using theoretical calculations and some experiments, which we have planned for the future.
Comment 2: The authors showed that silicon ether substrate gave up to 95% yields in the SuFEx reaction. The authors are asked to provide a plausible reaction mechanism and some mechanistic study of why the phenol substrate provided lower yields.
Response 2: The authors thank the reviewer for this important remark. Such high yields could be due to the formation of the strong silicon-fluorine bond during the reaction. However, the investigation of reaction mechanism requires a lot of time and resources and is beyond the scope of this manuscript.
Comment 3: Authors discussed the predicted pharmacokinetics and other ADMET properties of 1,4-naphthoquinones. Can some real ADMET properties and pharmacokinetics of a few potent compounds be provided to show the drug-likeness of the compounds?
Response 3: The experimental study of pharmacokinetics and ADMET properties, including the toxicological experiments in mice, will be undertaken and published in a separate paper.
Comment 4: In the experimental section, there is no mention of MHz in 1H NMR. Please provide. In the 13NMR, it is mentioned that 400 MHz. Is it performed on a 1600 MHz machine? Please correct it. NMRs are not written properly. The format is not correct. Please check it.
Response 4: The authors thank the reviewer for this important correction. The NMR operation frequencies in MHz are given in section 3.1. Reagents and Equipment: “The 1H and 13C NMR spectra were recorded on a Bruker AVANCE III HD instrument (operating frequencies: 1H - 400 MHz; 13C - 100 MHz).” According to the reviewer’s recommendation, we have checked and made necessary corrections of the NMR description format in the revised version of the manuscript.